# Improving Influenza Nomenclature Based on Transmission Dynamics

**DOI:** 10.3390/v17050633

**Published:** 2025-04-28

**Authors:** Jwee Chiek Er

**Affiliations:** Norwegian Veterinary Institute, 0454 Oslo, Norway; chiek.er@vetinst.no

**Keywords:** influenza A virus, nomenclature, transmission dynamics, zoonotic potential, molecular evolution, public health, virology

## Abstract

Influenza A viruses (IAVs) evolve rapidly, exhibit zoonotic potential, and frequently adapt to new hosts, often establishing long-term reservoirs. Despite advancements in genetic sequencing and phylogenetic classification, current influenza nomenclature systems remain static, failing to capture evolving epidemiological patterns. This rigidity has led to delays or misinterpretations in public health responses, economic disruptions, and confusion in scientific communication. The existing nomenclature does not adequately reflect real-time transmission dynamics or host adaptations, limiting its usefulness for public health management. The 2009 H1N1 pandemic exemplified these limitations, as it was mischaracterized as “swine flu” despite sustained human-to-human transmission and no direct pig-to-human transmission reported. This review proposes a real-time, transmission-informed nomenclature system that prioritizes host adaptation and sustained transmissibility (R_0_ > 1) to align influenza classification with epidemiological realities and risk management. Through case studies of H1N1pdm09, H5N1, and H7N9, alongside a historical overview of influenza naming, we demonstrate the advantages of integrating transmission dynamics into naming conventions. Adopting a real-time, transmission-informed approach will improve pandemic preparedness, strengthen global surveillance, and enhance influenza classification for scientists, policymakers, and public health agencies.

## 1. Introduction

Infectious disease nomenclature serves not only as a technical label for scientists but also as a critical tool in public communication and pandemic response. Historically, influenza A virus (IAV) naming conventions are based on viral characteristics like surface proteins or places of isolation, which fail to convey information about how the virus spreads or about host adaptations in real time. This disconnect can hinder public health messaging and response efforts. In this review, we examine the existing influenza nomenclature systems and their limitations and propose an improved framework centered on transmission dynamics. The review begins with an overview of IAV biology and the current naming scheme, followed by a historical perspective on influenza pandemic nomenclature. Core to this review is highlighting why transmission dynamics—such as the ability of a virus to sustain human-to-human transmission or cross species barriers—can augment information in virus names. Finally, we outline how a transmission-based nomenclature could be implemented, present supporting case studies, and offer recommendations for integrating this system into the global public health and surveillance infrastructure.

### 1.1. Influenza A Virus: An Overview

Influenza A viruses belong to the Orthomyxoviridae family and possess a segmented negative-sense RNA genome. The genome consists of 8 segments encoding at least 11 proteins, including the surface glycoproteins hemagglutinin (HA) and neuraminidase (NA), which form the basis of the current subtype classification. There are 19 HA subtypes (H1–H19) and 11 NA subtypes (N1–N11) known, theoretically yielding 209 possible HA/NA combinations, most of which are found in wild bird reservoirs [1,2]. Influenza A has a broad host range encompassing birds, various mammals (swine, horses, canines, marine mammals), and humans, reflecting its significant zoonotic and pandemic risk [3]. Two main processes drive viral evolution: antigenic drift (the accumulation of point mutations) and antigenic shift (reassortment of gene segments when a host has coinfection of multiple strains). These processes enable influenza viruses to evade immunity and occasionally jump between species [4]. Notably, many human influenza pandemics came from viruses originating or mixing in animal hosts [5]. Given this dynamic evolution and host range, a static naming convention fails or is slow to capture the properties of an influenza strain that are most relevant to its spread and control.

### 1.2. Challenges with the Current Nomenclature System

The prevailing influenza virus nomenclature was established by the World Health Organization (WHO) in 1953 and includes the following components for influenza A: antigenic type (A/B/C), host of origin (if not human), geographic location of isolation, strain number, year of isolation, and HA/NA subtype [6]. For example, A/California/07/2009 (H1N1) denotes a strain isolated in California in 2009 with H1N1 subtype. This standardized format has utility for basic identification and tracking in research. However, it exhibits several shortcomings in practice.

#### 1.2.1. Focus on Static Attributes

The WHO system emphasizes historical and genetic information (origin and subtype) but does not account for the dynamic epidemiological behavior of the virus. It captures where and from what species a virus was first isolated, but not how it spreads or adapts thereafter. Key factors such as real-time transmissibility and host adaptation are absent. For instance, genetic sequencing can identify a virus’s ancestral origin, but it does not convey whether the virus is currently spreading efficiently in humans or other hosts.

#### 1.2.2. Geographic Labels and Stigma

Names often include a location (e.g., “Asian flu”, “Spanish flu”), which may inadvertently stigmatize regions or populations without providing actionable information. A prominent example is the 2009 “swine flu” pandemic, initially being called “Mexican flu” by some media [7], causing diplomatic tensions and economic impacts on Mexico despite the virus’s widespread nature. Geographic names can mislead the public regarding the actual source or risk of a virus and may discourage transparency in reporting outbreaks.

#### 1.2.3. Host-Origin Labels and Cross-Species Complexity

Similarly, host-based labels (e.g., avian influenza, swine influenza) can be confusing when viruses jump species. Influenza viruses are often named for the species from which they were first isolated, but many strains do not remain exclusive to that host. For example, the A(H1N1)pdm09 virus contained gene segments from avian, swine, and human influenza lineages [8] yet established itself as a human virus. The PB1 gene of H1N1pdm09 originated in birds, but the virus now circulates predominantly in humans. Likewise, an “avian” H5N1 virus can infect people, and a “swine” virus can reassort with human strains. Rigid host labels fail to capture these transitions and can cause confusion; for instance, truly swine-adapted influenza strains (like endemic swine H3N2) [9] are distinct from the pandemic H1N1pdm09, yet both have been colloquially termed “swine flu”.

#### 1.2.4. Lack of Epidemiological Context

The current nomenclature does not indicate whether a virus is undergoing sustained human-to-human transmission or is causing only sporadic spillover infections. This distinction is crucial for public health. A name like A/H5N1 gives no indication that, as of now, H5N1 is chiefly a bird virus with rare human cases. Similarly, A/H7N9 conveys the subtype but not that H7N9 infected over 1500 humans via poultry exposure with limited, non-sustained human spread [10]. The absence of a transmission context in names is a missed opportunity to signal the level of pandemic risk associated with a strain.

Due to these challenges, the current naming system may hinder effective risk communication. Health authorities have often needed to introduce ad hoc nomenclature fixes. For example, during the 2009 H1N1 pandemic, the WHO avoided using geographic or species names and eventually standardized the term “A(H1N1)pdm09” (for “pandemic 2009”) to distinguish the novel human strain from endemic H1N1 viruses and to avoid the misleading term “swine flu” [11]. This change, implemented in 2011, required global coordination but came only after initial confusion. It highlights the need for a proactive naming framework that inherently carries epidemiological meaning, rather than retroactive corrections.

### 1.3. Historical Pandemic Nomenclature

Historical influenza pandemics illustrate how naming conventions have evolved with changing technology. We have reached a point where we need a new approach to the nomenclature of influenza to include epidemiology and transmission dynamics. In the pre-molecular era, we named pandemics after geographical locations or popular attributions, which bore little relation to the viruses’ true origin or characteristics.

#### 1.3.1. “Russian Flu” 1889–1890

The pandemic of 1889, one of the last great pandemics of the 19th century, was nicknamed the “Russian flu” (also “Asiatic flu”) because early reports came from St. Petersburg, Russia. This outbreak caused an estimated 1 million deaths worldwide [12]. There were debates over scientific evidence indicating the viral agent influenza A virus of the H3N8 subtype or a coronavirus [13]. Without virological information in 1889, the adoption of “Russian flu” presumed its geographic origin. Unsurprisingly, the label “Russian flu” did nothing to describe the virus’s behavior; indeed, the pandemic rapidly spread globally, regardless of its Russian association [12]. The use of a place name also risked stigma, although in this case, the term became primarily historical.

#### 1.3.2. “Spanish Flu” 1918–1920 (H1N1)

The infamous 1918 pandemic [14] was commonly called the “Spanish flu”, not because it originated in Spain (in fact, the first known cases were in the United States in early 1918), but because Spain, being neutral in World War I, openly reported on the outbreak, while warring nations censored news to maintain morale. The name “Spanish flu” thus reflected political circumstances rather than virology. This pandemic infected roughly one-third of the world’s population and killed an estimated 50 million people, making it the deadliest influenza pandemic on record [15]. The etiologic agent was later identified as an H1N1 influenza A virus. All eight genomic segments of the 1918 virus were eventually sequenced from preserved specimens, revealing an avian-derived influenza virus that adapted to humans [16]. However, in 1918, people had no conception of subtypes or transmission dynamics—the virus was unnamed, except by the misnomer. The term “Spanish flu” provided no useful information on how the virus spread or which populations were at risk. It also unfairly linked the disease to Spain. Modern analyses have since shown that the 1918 H1N1 virus became the progenitor of all later influenza A pandemics, but contemporary nomenclature did not capture any of this critical information.

#### 1.3.3. “Asian Flu” 1957–1958 (H2N2)

The 1957 pandemic was caused by a novel H2N2 virus that arose from reassortments between an avian influenza and the previously circulating human H1N1 strain [17]. It was termed the “Asian flu” because it was first identified in East Asia (with early outbreaks in China and Hong Kong). This pandemic was milder than that of 1918 but still caused an estimated 1–2 million deaths worldwide. Virologists at the time were able to identify the new H2N2 subtype using serological methods, a significant advance in influenza science. Thus, the virus was also called “Asian H2N2” [17]. While the subtype indicated a new antigen against which few people had immunity, the geographic moniker “Asian” again did not convey the shift of introducing an avian HA and NA that made this pandemic possible. However, it may have contributed to bias or complacency outside of Asia, despite the virus spreading globally in a matter of months. The naming did nothing to highlight the virus’s high human transmissibility, which is the key reason it became a pandemic.

#### 1.3.4. “Hong Kong Flu” 1968–1970 (H3N2)

The 1968 pandemic, caused by an H3N2 virus, was named after Hong Kong, where the virus was first reported in July 1968 [18]. The H3N2 strain emerged through reassortment of the 1957 H2N2 virus with an avian virus (introducing a new H3 HA gene while retaining the N2) [19]. The “Hong Kong flu” caused around 1 million deaths worldwide (some estimates range up to 4 million) and was the least severe of the 20th century pandemics. By 1968, the science of influenza had advanced enough that the subtype H3N2 by antigenic surface coat identified the influenza virus (often referred to as “Hong Kong H3N2”). However, public communications still centered on the geographic label. Associating the disease with Hong Kong potentially stigmatized that region and gave a false impression that the threat was localized [20]. In reality, the virus spread internationally within weeks, reaching the United States by that same year. The H3N2 virus became established as a human seasonal strain (replacing H2N2) and continues to circulate today [19]. Knowledge of its efficient human transmission and mild-to-moderate virulence was far more relevant to public health than its geographic origin, yet the name did not reflect those aspects.

These historical examples show a pattern of influenza pandemics being named in ways that emphasize origin or locale (Spanish, Asian, Hong Kong, Russian) rather than the viruses’ transmissibility or host adaptation [21]. Such names often arose colloquially and stuck due to convenience, but they lack scientific precision and can mislead or stigmatize. In the modern era, virologists do assign subtype designations (e.g., H1N1, H3N2) as was done in 1957 and 1968, which is a step forward. However, subtype alone still conveys limited information about a strain’s current epidemiological behavior. For instance, there have been multiple distinct H1N1 influenza pandemics or epidemics (1918, 1977, and 2009) and numerous H3N2 variants; the subtype label does not distinguish these in terms of transmission dynamics [22]. 

### 1.4. Understanding Transmission Dynamics and R_0_

In epidemiology, a central concept for understanding how infectious diseases spread is the basic reproduction number, denoted R_0_ (R-naught) [23]. R_0_ represents the average number of secondary infections produced by a single infected individual in a fully susceptible population [24]. If R_0_ is greater than 1, an infection can spread sustainably within the host population, indicating the potential for an outbreak or epidemic [25]. Conversely, if R_0_ is less than 1, transmission is self-limiting, and the infection typically dies out without sustained spread. For influenza viruses, estimating R_0_ helps distinguish between isolated zoonotic spillovers (e.g., avian influenza infecting a human from poultry) and viruses capable of driving epidemics or pandemics through sustained human-to-human transmission. Incorporating R_0_ and transmission thresholds into influenza naming would allow better virus classification to reflect the actual epidemiological risk posed to humans and other species.

#### Need for a New Approach

The inadequacies of historical nomenclature have been evident in recent years. WHO and other bodies now discourage the use of geographic names for new pathogens to avoid stigma (as seen with the naming of COVID-19 variants Alpha, Beta, etc., instead of with country names) [26]. Influenza strain naming, however, remains largely rooted in the mid-20th-century convention. Given our improved understanding of influenza ecology, evolution, and the One Health context (the intersection of human, animal, and environmental health), it is timely to refine influenza nomenclature. In particular, incorporating transmission dynamics and host adaptation status into virus names could provide immediate insight into the public health risk a strain poses. In the sections that follow, we explore how the current classification systems for influenza operate, why they fail to capture these dynamic properties, and how a transmission-based naming framework can address these gaps.

## 2. Existing Influenza Virus Classification Systems

Current influenza classification and naming conventions can be characterized broadly by (a) phenotypic subtype classification based on surface proteins, and (b) host-based classification. Both are used often in scientific and public health communications, often alongside the standardized influenza strain-naming format described earlier. We review these approaches here to highlight their scope and limitations.

### 2.1. Surface Protein Classification

The accepted convention of classifying influenza A viruses is by their surface glycoproteins: hemagglutinin (H or HA) and neuraminidase (N or NA). As noted, there are multiple HA and NA subtypes, and an isolate is typically referred to by the combination of these, such as H1N1, H3N2, H5N1, etc. This system is useful because the HA and NA are major antigenic determinants of a population’s immunity, and vaccines are designed according to these subtypes. For example, the seasonal flu vaccine might include an H1N1 and an H3N2 strain referring to the HA/NA of circulating human viruses [27].

Subtype nomenclature readily conveys certain information, especially if one or few hosts are involved. For instance, subtype H5N1 immediately is associated with highly pathogenic avian influenza in birds (and occasional human infections), whereas H3N2 has been associated with milder, sustained human circulation since 1968. Subtyping also facilitates rapid risk assessment when novel strains emerge; the appearance of an H7 or H5 in humans triggers concern due to its known virulence in birds [28].

However, subtype classification alone has limitations. There are 19 HA [2] and 11 NA subtypes identified, mostly in birds, and theoretically 209 combinations, many undiscovered in nature, and conversely, viruses of the same subtype can vary greatly in host range and virulence. A case in point is H1N1: sharing this same subtype are the 1918 pandemic virus, the 2009 pandemic virus, and various swine and avian strains, a range of viruses sharing the same H1N1 subtype but with very different behaviors. Similarly, H3N2 could refer to human seasonal flu or an avian strain in ducks. Subtyping does not capture these distinctions [1,29].

The subtype naming can also lead to public confusion. The term “H1N1” became infamous during the 2009 pandemic (labeled “swine flu”); even today in 2025, 16 years after its emergence in 2009, humans are still the dominant, persistent host, showing sustainable transmissions, and are the reservoir hosts of H1N1pdm09 [30]. Yet there were earlier H1N1 viruses circulating in humans long before 2009, and they continue to do so. In fact, after 2009, both the old seasonal H1N1 (pre-2009 lineage) and the new pandemic H1N1 co-circulated for a time, which required researchers to differentiate them as H1N1pdm09 vs. H1N1 (seasonal) [31]. The average person hearing “H1N1” cannot differentiate the strains of H1N1 in circulation.

Another element of subtype-centric naming is the pathogenicity designation in avian hosts. H5 and H7 avian influenza viruses are further categorized as highly pathogenic (HPAI) or low pathogenic (LPAI) based on molecular traits and chicken lethality [32]. For example, H5N1 is usually assumed to be HPAI (highly pathogenic avian influenza) when discussed. Yet this too is context-specific: H5N1 in a wild bird reservoir [33] might be low pathogenic until it mutates in chickens. Hence, the name “H5N1” by itself does not indicate pathogenicity or spread.

In summary, surface protein classification (H or N) is a necessary component of influenza nomenclature—maintained as a foundational layer in our proposed system—but it addresses primarily the viral antigenic identity (for immunity and virology) rather than the transmission or host adaptation status. Our review article presents the arguments that combining subtype information with transmission dynamics in nomenclature could greatly enhance the utility of virus names.

### 2.2. Host-Based Classification

Another traditional way to label influenza viruses is by their host of origin. Terms like “avian influenza,” “swine influenza,” “equine influenza,” and “human influenza” use host names to denote the species populations in which a virus is currently circulating [34]. This classification stems from the ecology of influenza A: certain subtypes are associated with particular hosts (for instance, H3N8 in horses [35,36] and dogs [37], H1N1 and H3N2 in humans and swine, etc.) [38]. In the research literature and surveillance reports, one often sees tags such as LPAI H7N9 (avian) [39] or variant H3N2 (swine-origin) [40].

While host labels can be convenient shorthand, they are frequently messy in practice due to cross-species transmission. Influenza viruses do not always respect species boundaries [41]. Swine, for example, are susceptible to avian and human influenza strains and can serve as “mixing vessels” for reassortment. A virus might originate in one species and then adapt to another. The current naming system accounts for the initial host in the strain name (e.g., A/duck/…, A/swine/… if not human), but once a virus becomes established in a new host, the original host-based name may become misleading, because a new reservoir host is established, and dissemination of the virus will come from the new reservoir host. 

A clear illustration is the 2009 H1N1 pandemic virus. It was the product of reassortment among influenza viruses from swine, birds, and humans, and it likely had been evolving in pigs for years before spilling over to humans. However, it was first dubbed a “swine flu” because genetic analysis showed swine lineage origins, and initial media reports tied it to pig farms, even though right from the start, the virus was isolated from humans, and it was spreading widely in humans and became a predominantly human virus [42]. Calling it a swine flu strain after mid-2009 was inaccurate in terms of ongoing transmission—by then, it was a human pandemic strain [43,44]. The host-based label stuck in the public consciousness, however, and caused significant issues: some countries banned pork products and unjustifiably culled pigs, and people were confused about food safety, even though the virus was being transmitted person-to-person [45,46]. The WHO’s adoption of the term A(H1N1)pdm09 was partly to rectify this issue [11,47,48]. This example underscores how a virus can leap from one host to another, rendering the original host label obsolete or even harmful for communication.

Another example is H5N1 avian influenza. H5N1 was first recognized in 1997 in Hong Kong’s poultry and human infections [47,48], and it re-emerged in 2003 to spread epizootically in birds across Eurasia and Africa. It was called “bird flu” because it primarily affected birds and poultry. However, H5N1 has infected hundreds of humans (often with severe outcomes), as well as various mammalian species (cats, tigers, seals, etc.). In recent years, the H5N1 clade 2.3.4.4b virus has caused infections in a wide range of mammals, including outbreaks in farmed mink and wild mammals [49]. If H5N1 were to mutate to sustained transmission in mammals or humans, the label “avian influenza” would no longer be appropriate. We would need to signal quickly that the virus is no longer just a bird threat. Current practice would be to note this in prose (e.g., “H5N1 adapted to mammals”), but the nomenclature itself does not change unless a completely new subtype or strain name is assigned.

Host-based naming also struggles with influenza viruses that circulate in multiple species simultaneously. Influenza A typically has reservoirs in wild birds [50], but many strains have jumped to domestic poultry, swine, horses, dogs, etc. [51,52]. For instance, an H3N2 lineage that originated in birds jumped to pigs and humans (the 1968 pandemic strain in humans, which later even jumped to dogs) [20]. Similarly, certain H1N1 and H3N2 viruses circulate endemically in swineherds and occasionally infect humans (termed variant viruses, like H1N1v or H3N2v). In those cases, what do we call the virus? If it is in pigs, we say swine H3N2, if it infects a human, we say variant H3N2, but genetically, it can be nearly the same virus. The label changes with the host context, which can be confusing if not carefully explained.

In summary, host-of-origin terminology is a double-edged sword: it highlights the ecological source of a virus, which is important for veterinary and One Health contexts, but it can become misleading if the virus expands beyond that host. A more dynamic naming system would update the host descriptor when a virus establishes sustained transmission in a new host. Our transmission-based nomenclature proposed in Section 4 aims to do exactly that—e.g., an “H5N1-Avian” could become “H5N1-Mammal” if evidence of sustained mammalian spread emerges. This way, the name itself keeps track of the virus’s host adaptation status.

Before detailing the proposal, we first summarize the key limitations of the current nomenclature approach, which motivate the need for change.

## 3. Limitations of the Current Nomenclature System

Under the status quo, influenza nomenclature provides static labels that often fail to convey epidemiologically significant information. The main shortcomings are as follows: (1) static attributes emphasized over dynamic behavior, (2) potentially misleading host labels, and (3) public health communication challenges.

### 3.1. Static Attributes

The current system labels viruses based on their genetic subtype and initial isolation context, which remain fixed even as the virus spreads and evolves. This static approach neglects the changing transmission dynamics and phenotypic behavior of the virus. For instance:

#### 3.1.1. Geographic and Historical Emphasis

The inclusion of a place name (e.g., “Asian flu” for H2N2, “Spanish flu” for H1N1) provides a historical origin, but inaccurate as they may have been, more importantly, they lacked information on ongoing situational insight. Such labels can even be counterproductive, as they may stigmatize regions without offering actionable data about the virus’s current threat level. A person hearing “Asian flu” learns nothing about how contagious or virulent the virus is or whom it is primarily affecting.

#### 3.1.2. Genetic Ancestry vs. Current Behavior

Modern strain names often incorporate clade or lineage markers (for example, “A(H5N1) clade 2.3.4.4b” or “A/California/07/2009(H1N1)”), which reflect genetic relationships. While valuable for scientists tracing evolution, these designations do not indicate whether the virus is spreading efficiently in humans or other hosts at that time point [53]. A name locked to genetic identity can quickly become outdated in meaning. Though H7N9 in 2013 was an avian-origin virus, five years later, thousands of human infections were recorded, mostly from widespread poultry-to-human transmission. The genetic label remained the same, even as public health concerns grew.

#### 3.1.3. No Temporal Component

Other than the strain isolation year (used in full strain names), there is typically no indication of time in influenza naming. Thus, the nomenclature does not readily distinguish a 2009-era “swine flu” virus from a 2019 descendant of the same lineage. In contrast, epidemiologists often have to introduce terms like “seasonal H1N1 2015” vs. “pandemic H1N1 2009” to clarify. A more dynamic system might include temporal markers to denote when a major host-transition event occurred (e.g., adding the year a virus adapted to humans).

Influenza viruses are moving targets—they mutate, reassort, and adapt, like H5N1 adapting to dairy cattle and mink and many different mammals [54]. A static name captures them like a snapshot, potentially missing the “movie” of their spread. This can lull observers into a false sense of familiarity (“oh, it is just H5N1, we have known that for years”) when, in reality, the risk profile of a virus may change if it breaches new species barriers or gains transmissibility. The nomenclature needs to be more fluid to reflect such shifts.

### 3.2. Misleading Host Labels

As discussed, naming by host origin can be problematic when a virus jumps hosts. The nomenclature often fails to distinguish between a virus’s historical origin and its current main host, leading to confusion.

#### 3.2.1. Persistence of Origin Labels

The 2009 H1N1 example shows how the “swine flu” label persisted even after the virus became a human pandemic strain. Similarly, the influenza research community still refers to the 1918 virus as avian-like H1N1, yet in 1918, it was obviously a human epidemic. The label emphasizes where it came from (historical host, birds), not where it is (current host, people). In day-to-day public health usage, we do not give it an undated name “human-adapted H1N1”—it was retrospective in research papers. This lag in language can affect how seriously the public and policymakers take emerging infections. If H5N1 were to start spreading easily among people, continuing to call it “avian influenza H5N1” might downplay the new reality that it has evolved into a human epidemic.

#### 3.2.2. Conflation of Distinct Strains

Host labels can cause the conflation of different viruses. For example, H1N1 in swine (classic swine flu) versus H1N1pdm09 in humans are distinct lineages [55], but the shared “swine” association caused public misunderstanding. Another case is the term “bird flu” applied to H5N1, H7N9, and other avian strains. If someone says “bird flu outbreak,” it could mean very different viruses with different human risk levels. Yet the host-oriented name obscures the distinction. During 2013–2017, China experienced hundreds of human infections with H7N9 avian influenza [56], a situation quite distinct from the H5N1 bird flu in other countries. The layman will not grasp this nuance from names alone.

#### 3.2.3. Zoonotic vs. Sustained Transmission

It is critical to distinguish real pandemic risk by whether the virus is merely zoonotic to an individual or has achieved sustained human-to-human transmission. The current nomenclature does not encode this difference. For instance, both H5N1 and H1N1pdm09 at times have been called “swine influenza” or “avian influenza” in broad terms, but H1N1pdm09 achieved human transmission (R_0_ > 1 in humans), whereas H5N1 has not (only rare clusters). Without additional explanation, the public might think “bird flu” in 2006 (H5N1) and “bird flu” in 2013 (H7N9) were similar, yet sustained human-to-human transmission was not the case but was largely a zoonotic epidemic via humans visiting live poultry markets [56]. A naming scheme that explicitly tags a virus as, say, “H7N9-Avian” indicates its primary host context (birds). Hypothetically, if H7N9 begin spreading between humans, it might be renamed “H7N9-Human”. Such clarity is missing in our current, static labels.

### 3.3. Public Health Communication Challenges

Because of the above issues, the current nomenclature can impede effective communication and timely response.

#### 3.3.1. Stigma and Concealment

Countries might hesitate to report novel viruses if they fear being permanently associated with a pandemic (the “Spanish flu” effect). A neutral, behavior-based naming system could reduce this disincentive by not embedding location in the public name. Instead of “Country X flu,” a name could focus on the host/transmission (e.g., “avian-to-human flu 20XX”), which is less stigmatizing and more descriptive of the situation. Transparency in reporting outbreaks is essential, and naming conventions [57] should facilitate rather than hinder it.

#### 3.3.2. Public Misunderstanding

The public often has limited scientific knowledge of influenza. If names are misleading, people may underestimate or overestimate the threat. For example, the term “low pathogenic avian influenza” (LPAI) refers to pathogenicity in birds, but a strain that is LPAI (mild in poultry) can still infect and severely harm humans (as H7N9 did). Someone might falsely assume “low pathogenic” means it is not dangerous to anyone. Likewise, during the 2009 pandemic, some avoided pork products due to the term “swine flu” despite no risk from properly cooked pork—a costly misunderstanding for agriculture. A well-designed nomenclature could improve risk communication by immediately conveying the nature of spread (e.g., “human-transmissible flu strain”) rather than a possibly irrelevant origin.

#### 3.3.3. Policy and Response Delay

Public health measures often rely on clear recognition of a virus’s status. If the name does not make it obvious that a virus has switched from animals to people, officials might delay interventions. Consider how quickly we would react to reports of “sustained human H5N1 influenza” versus just “H5N1 avian influenza cases”. The former phrasing (which a transmission-based nomenclature would provide inherently) triggers a sense of urgency. The latter might sound like the familiar sporadic bird-to-human events we have seen for years. Thus, a dynamic naming system could prompt swifter response when a virus meets the criteria for higher pandemic risk.

In summary, current influenza naming conventions, while deeply entrenched and useful for virological documentation, suffer from inflexibility and a lack of epidemiological context. These limitations can obscure critical changes in a virus’s behavior and impede clear communication. The following section introduces a proposed framework to address these issues by incorporating transmission dynamics into the nomenclature.

## 4. Proposed Framework: Transmission-Based Nomenclature

To enhance influenza nomenclature, we propose a framework that retains the essential virological information (such as subtype and origin) but adds transmission dynamics and host adaptation status as key elements of the name. The goal is a system wherein the name of the virus evolves in tandem with the virus’s demonstrated behavior, especially with regard to which host(s) are sustaining transmission.

### 4.1. Core Principles of Transmission Dynamics-Driven Nomenclature

The transmission-based nomenclature framework rests on four core principles:

1. Dominant Host Adaptation: The naming should reflect the current primary host in which the virus is spreading sustainably. In practical terms, this means identifying the host species or category where R_0_ > 1 (the virus can maintain chains of transmission). For example, if an H5N1 virus is at first circulating in birds (R_0_ > 1 in bird populations) and only sporadically infecting humans (R_0_ ~0 in humans), it would be labeled with an avian marker. If it later adapts to enable efficient human-to-human transmission (R_0_ > 1 in humans), the host marker in the name would switch to human. This principle ensures the name always points to the epidemiologically most relevant host. It is important to define host categories broadly (human, avian, swine, equine, etc., or potentially, “mammalian” for non-human mammals) for simplicity. The criteria for name change triggers state that only sustained transmission in a new host (R_0_ > 1) or a fundamental shift in host adaptation would warrant a nomenclature update. This change will not occur for incidental spillovers or minor mutations [58,59].

2. Dynamic Updates: Updating host names would happen only upon significant changes in transmission dynamics, not minor epidemiological events. In other words, the nomenclature changes are triggered by clear evidence of sustained transmission in a new host or a fundamental change in virus behavior, rather than every small cluster or mutation. This avoids instability in naming. Once a virus establishes a self-sustaining transmission cycle in a new host population, it merits a naming update. However, the original name and classification are retained in the scientific records for continuity. An updated name signals to public health, “this is effectively a new phase” for the virus. If multiple hosts maintain the virus in parallel (e.g., some influenza strains co-circulate in pigs and humans), the naming could include both or default to the host representing the greatest public health concern (perhaps using a hierarchy like human > other mammals > birds, since a human-adapted virus usually implies the highest pandemic risk).

3. Integration with Genetic Data: The scheme would retain the subtype and other genetic identifiers as needed, but these would not be the sole focus of the public-facing name. Genetic nomenclature (like clade numbers or lineage names) can be appended or kept in parenthetical notation for scientists. For example, a full name might be “H7N9-Avian (Yangtze River Delta lineage)” to satisfy both needs. Crucially, however, the public or headline name would be the transmission-based one, emphasizing that H7N9 is currently an avian virus, while the detailed genetic information is secondary. This integration ensures that valuable molecular data are not lost; researchers can still trace origins and relatedness. However, by not prioritizing genetic lineage in the primary name, we reduce confusion for policymakers and the public, focusing them on what matters for control (how and where the virus is spreading now).

4. Transparency and Clarity: The naming system should incorporate temporal markers or neutral geographic markers, when relevant to track epidemiology, but do so in a way that avoids blaming specific regions. For instance, using the year of emergence or host switch can provide a time reference (as with “pdm09” for the 2009 pandemic). If needed, broad geographic terms might be used without stigma—e.g., “H5N1-Mammal-Europe2022” could denote a mammalian-adapted H5N1 first noted in Europe in 2022, without singling out a country. However, any geographic element would be included only for context and not as a core part of the virus identity. The priority is clarity in what the virus is doing. Thus, the naming framework strives to be unambiguous and easily interpretable: anyone reading the name should glean the subtype, the primary host of spread, and possibly a timeframe indicator, all of which are directly relevant to risk assessment.

These principles collectively aim to produce names that are scientifically sound yet adaptable. The system acts almost like an annotation of the virus’s status. As the virus evolves, so does its nomenclature classification. This is analogous to how storm naming can escalate (tropical depression to tropical storm to hurricane categories), signaling the changing status, except here the stakes are in virological and epidemiological terms. In certain cases, the timing and coverage of vaccination campaigns may intersect with viral evolution, particularly when vaccine escape variants emerge and lead to sustained outbreaks in previously immune populations [60,61]. While vaccination is not a naming criterion per se, it may influence the interpretation of transmission dynamics and, therefore, the decision to update a virus’s host-associated nomenclature.

### 4.2. Illustrative Implementation of Transmission-Based Naming

To make the proposed framework more concrete, we provide examples of how it could be applied to known influenza A viruses. These examples demonstrate the naming conventions and the conditions under which names would update.

#### 4.2.1. Example 1: H1N1pdm09 (2009 Pandemic Strain) Initial Name

When the virus emerged in 2009, naming by transmission dynamics would have appropriately pointed to a name “H1N1-Human-2009”, or simply, “H1N1-Human (2009)”, reflecting that it was a novel H1N1 sustaining human-to-human transmission starting in 2009 in the human population. This emphasizes the virus’s pandemic spread in humans. Indeed, the virus quickly became the dominant human seasonal flu strain after 2009. Rationale: The virus demonstrated R_0_ > 1 in humans almost immediately in multiple countries. Under our system, the host descriptor, “Human”, tells us this is a human-transmissible strain. The inclusion of “2009” as a temporal marker distinguishes it from other H1N1 variants. In practice, WHO named it A(H1N1)pdm09, which is very similar in concept (subtype + pandemic year tag). Our system formalizes this kind of designation. Updated name: If we consider its status today, it is still an H1N1 predominantly in humans (as a seasonal virus), so it would remain H1N1-Human in the nomenclature. Should it, hypothetically, jump back into animals and form sustained lineages there distinct from humans, we might then label those separately (e.g., H1N1-Swine-201x) for a lineage derived from the 2009 virus now established in pigs, which indeed has happened to some degree, though often, those are referred to as variant swine viruses [62].

#### 4.2.2. Example 2: H5N1 Highly Pathogenic Avian Influenza—Initial Name: “H5N1-Avian”

For decades since its emergence, H5N1 has primarily circulated in birds, so the name conveys that the virus’s dominant host is avian. This covers the situation up to present, where H5N1 is causing a global panzootic in wild birds and poultry. Trigger for update: If a particular H5N1 clade acquired the ability for sustained mammalian transmission (for instance, an H5N1 strain spreading efficiently among mink [63] or an outbreak in mammals with clear mammal-to-mammal contagion), the name could be updated to “H5N1-Mammal” (with an optional year or clade notation, e.g., H5N1-Mammal-2024). This would immediately signal that H5N1 is no longer just a bird problem. If, furthermore, it started spreading among humans (and met the criteria for human adaptation), it would change to “H5N1-Human”. Not every single mammalian case would prompt a change—only evidence of sustained transmission (R_0_ >1) in that new host. For example, the large H5N1 outbreak in a Spanish mink farm in 2022 suggested some mink-to-mink transmission [63], but whether it was sustained or just one farm event is still under study. One might wait for confirmation of onward spread beyond one cluster. On the other hand, the ongoing spread of H5N1 in wild mammals (e.g., among seals in multiple colonies) [64,65] and domestic dairy cattle [66,67] might prompt a “Mammal” designation if shown to be widespread. Outcome: At the time of writing, H5N1 remains principally avian, so it stays H5N1-Avian. If a change happens, the year of that shift could be appended (e.g., H5N1-Mammal-2023 if that was the year it first sustained mammalian transmission). Genetic clade info (like 2.3.4.4b) might be mentioned in scientific contexts but not in the primary name.

#### 4.2.3. Example 3: H7N9 Avian Influenza—Initial Name: “H7N9-Avian”

When the H7N9 virus was first detected in humans in 2013 in China, it was traced to poultry sources [56,68]. Through 2017, H7N9 caused 5 epidemic waves in humans, totaling 1568 laboratory-confirmed cases, with about 39% mortality. However, nearly all those cases were due to repeated spillovers from infected chickens at live bird markets, not sustained human transmission. Under our framework, throughout that period H7N9 would retain the “-Avian” label because its propagation depended on birds (closing live poultry markets sharply reduced cases). Trigger for update: If at any point H7N9 had evolved to spread easily from human to human (signs of community transmission, clusters beyond households, etc.), the name would have switched to “H7N9-Human”. In reality, that did not occur; instead, H7N9 was controlled largely by poultry vaccination in 2017, and human cases dwindled. Therefore, H7N9 remains an avian virus in nomenclature. If it re-emerges or if any limited human-to-human spread is observed, one might consider an intermediate label like “H7N9-Zoonotic” or “H7N9-Avian?” to denote uncertainty, but that adds complexity. Its likely better to stick to confirmed sustained transmission events for renaming triggers.

These examples show that the naming would include the subtype (HxNy), a host marker (Human, Avian, Swine, etc.), and optionally, a time or episode marker (year of emergence or host jump). The subtype informs about the virology, the host marker tells the current epidemiology, and the time marker can differentiate separate emergences of what might be the same subtype. For instance, “H1N1-Human-1918” vs. “H1N1-Human-2009” could denote the two different H1N1 pandemics, which have distinct lineages, but the public name makes clear they are separate events.

Under this system, the formal strain names (with exact strain IDs) remain for lab records, but public communications and high-level discussions refer to the transmission-based names. The framework can also extend to influenza B (which infects only humans and seals—one could label lineages as Human or Seal if ever needed) and potentially to other emerging viruses for consistency.

By adopting transmission-based nomenclature, we maintain clarity as viruses evolve. It is a proactive approach: rather than catching up with the virus (as happened when naming “pdm09” after the fact), the naming framework is built to update in step with key epidemiological shifts. This could significantly improve how scientists, officials, and the public perceive and respond to emerging influenza threats.

## 5. Case Studies Supporting the Proposed Framework

To further illustrate and validate the proposed transmission-based nomenclature, we examine three case studies in detail: the 2009 H1N1 pandemic (H1N1pdm09), the persistent avian influenza H5N1, and the zoonotic H7N9 virus in China. Each case demonstrates particular challenges of the current naming system and shows how a transmission-focused name could provide clearer insight. We also draw on data and events associated with these viruses to underscore the importance of nomenclature that reflects transmission dynamics.

### 5.1. H1N1pdm09 (2009 “Swine Flu” Pandemic)

Background: The 2009 pandemic H1N1 virus emerged in April 2009 and swept across the globe within months [38]. Genetic analysis revealed an extraordinary origin: it was a quadruple-reassortant virus combining gene segments from North American swine, Eurasian swine, avian, and human influenza lineages. Specifically, its HA, NP, and NS genes traced to a classical swine H1N1 lineage; its PB2 and PA genes to North American avian influenza; its PB1 gene to a human H3N2 virus (which itself had come from birds originally); and its NA and M genes to a Eurasian swine H1N1 lineage. This mosaic had been incubating in pig populations (likely in Mexico) for years before spilling over to humans in around 2009 [69].

Even upon its initial detection in humans, the 2009 H1N1 virus was widely referred to as “swine flu” in the media, owing to its genomic segments originating from swine influenza lineages. However, within weeks it was clear that the virus was transmitting efficiently human-to-human in many countries—it had become a human-adapted strain. The continued use of “swine flu” caused public confusion and significant economic damage to the pork industry, as discussed. In terms of epidemiology, H1N1pdm09 caused a pandemic with an estimated ~284,000 deaths in the first year (based on later CDC estimates, considering many deaths were not lab confirmed), and it disproportionately affected children and non-elderly adults [70]. By August 2010, WHO declared the pandemic over, and the virus had become endemic as part of seasonal influenza. It is still circulating as one of the seasonal flu A strains to this day.

Nomenclature Issues: Initially, formal communications used “Influenza A (H1N1) 2009” and later “A(H1N1)pdm09”. These are technical and take time for adoption by the lay media, which stuck with “swine flu”. The host-based label misled some into thinking the virus came directly from pigs at the point of infection (without concrete cases; almost immediately, human-to-human transmission dominated). In addition, once it became a seasonal strain post-2010, calling it “H1N1pdm09 pandemic” was no longer appropriate, but simply calling it “H1N1” could mix it up with other H1N1 viruses. 

#### How a Transmission-Based Name Helps

Under our framework, as soon as it was determined that sustained human transmission was occurring (which was evident by May–June 2009), the virus would be labeled H1N1-Human-2009 (or abbreviated for the media as “2009 Human H1N1”). This name directly conveys that it is an H1N1 influenza virus adapted to humans as of 2009. It avoids the term “swine” entirely, removing the implication about pigs, and instead highlights the important fact: it is now a human epidemic. The year tag “2009” distinguishes it from any previous human H1N1 lineage. As the virus persisted in humans beyond 2009, the name H1N1-Human would still apply, perhaps with the year dropped after some time, when it is clearly the endemic strain. If discussing historical context, one could include the year for clarity (e.g., “the post-2009 H1N1-Human virus”).

This naming would have been useful when, for example, scientists later discussed antigenic drift of the “pandemic H1N1” virus in following years—they could simply say the Human H1N1 lineage. In fact, today we have two influenza A subtypes in seasonal circulation: H1N1 and H3N2, both in humans. Calling them H1N1-Human and H3N2-Human in a formalized way might make it clearer that these are human-adapted strains, as opposed to the many H1N1 or H3N2 variants in animals, pigs especially.

To summarize this case: H1N1pdm09 demonstrates that once a virus makes the jump to sustained human transmission, the nomenclature should promptly reflect that shift. A transmission-based name would have improved public understanding and reduced misnomers. It also highlights that such a name can continue to be relevant long after the initial event (as that virus becomes the new seasonal lineage).

### 5.2. H5N1 (“Avian Influenza” with Pandemic Potential)

Background: Influenza A(H5N1) first gained worldwide attention in 1997, when an H5N1 highly pathogenic avian influenza (HPAI) virus in Hong Kong infected 18 people, killing 6 [71,72]. After aggressive culling ended that outbreak, H5N1 resurfaced in 2003 and spread epizootically across Asia, Europe, and Africa, becoming entrenched in poultry in many countries. From 2003 through 2021, the WHO documented 863 human H5N1 cases with 455 deaths, an exceptionally high average case fatality rate (~53%) [54]. Nearly all these cases were due to direct contact with infected birds—human-to-human transmission was extremely rare and self-limited. H5N1 thus became the poster child of a deadly zoonotic virus that had not yet achieved person-to-person spread. Scientists and public health officials have long been concerned that H5N1 could mutate or re-assort to gain transmissibility among humans, potentially causing a severe pandemic.

In late 2021, a new chapter in the evolution of H5N1 began with the global spread of clade 2.3.4.4b, which had been circulating in birds. This clade triggered a panzootic—a pandemic-scale epizootic—causing unprecedented outbreaks among wild birds and domestic poultry across multiple continents [49]. As viral circulation intensified, spillovers into mammals became increasingly frequent. Since 2022, H5N1 infections have been documented in a wide range of mammalian wildlife—including foxes, mink, skunks, dolphins, bears, and seals—most likely through predation or scavenging of infected birds [73].

A turning point came in late 2022 with an outbreak at a Spanish mink farm, where evidence suggested mink-to-mink transmission following the initial introduction. Similarly, in early 2023, Peru reported over 600 sea lion deaths linked to H5N1 [74], raising further concerns about potential mammal-to-mammal transmission in densely populated colonies. These events highlight the virus’s growing ecological flexibility.

Moreover, viral detection in hosts like mink and sea lions may involve intra-host viral competition, quasi-species dynamics, and possible co-infections, complicating the precise attribution of pathogenesis and transmission chains [75]. In a surprising development, the detection of H5N1 in dairy cattle in the United States in early 2024 was emerging evidence of cow-to-cow transmission—an unprecedented shift for a virus classically confined to avian hosts.

While these developments have not yet resulted in sustained human-to-human transmission, they collectively signal that H5N1 is actively expanding its host range, posing new challenges for surveillance and preparedness.

Human cases of H5N1 remain very sporadic; a handful of cases (often, single cases or small family clusters) occurred in 2022–2023 in Asia and one in 2023 in South America (Ecuador), mostly with high-risk exposures to birds [76,77]. There has been, however, no evidence of sustained human spread to date [78]. The consensus so far is the risk to the public is low, but the situation is monitored closely by organizations like CDC and WHO.

Nomenclature issue: H5N1 has traditionally—and appropriately—been termed an “avian influenza virus”, reflecting its origin and predominant maintenance in avian reservoirs. However, the recent and widespread spillover of clade 2.3.4.4b into a diverse array of mammalian hosts, including isolated human cases, challenges the sufficiency of this designation [79]. Continued use of a static “avian flu” label may inadvertently obscure the expanding ecological range and zoonotic potential of the virus. To improve risk communication and surveillance sensitivity, the nomenclature should be responsive to host transitions and reflect significant shifts in transmission dynamics, particularly when evidence of mammalian adaptation surfaces.

#### How a Transmission-Based Name Helps

In a transmission-based framework, as of 2023 we would label the circulating strain as H5N1-Avian (since birds are the sustaining host). We would continue to monitor mammalian clusters closely. The moment evidence meets a defined threshold for sustained mammal-to-mammal transmission (which could be a certain number of non-contained outbreaks among mammals or a detected R_0_ > 1 in a mammalian species), we would update the name to H5N1-Mammal (Year) for that lineage. For instance, the outbreaks in mink and sea lions in 2022–2023, if confirmed as onward transmission, could justify an “H5N1-Mammal-2022” designation for that specific context. If the virus then adapts further to humans (a scenario we fear but have not seen), any sustained human transmission would trigger a renaming to H5N1-Human.

Such naming would make communications more precise. Officials could say, “We now have H5N1-Mammal strains identified, which signals the virus has adapted to at least some mammalian species—we are escalating preparedness”. This could correspond, for example, to seed strain development for human vaccines. If, unfortunately, H5N1 ever shows person-to-person spread in a community, calling it H5N1-Human immediately frames it as a pandemic threat requiring aggressive containment. It would distinguish that scenario from the decades of referring to H5N1 as a zoonotic avian flu. Additionally, by encoding the year of adaptation, it would help researchers track evolutionary events. (It is conceivable that multiple distinct H5N1 mammalian adaptations could occur in different places/times—each might be given its own label, like H5N1-Mammal-2024A vs. 2024B if needed).

In the meantime, while H5N1 remains an avian virus with only sporadic spillover, the name H5N1-Avian reinforces that the primary risk is via contact with infected birds. It also hints at the control strategy—controlling it in birds is key. A name change is a red flag that our priorities must shift to containing it among mammals or humans. This aligns nomenclature with action.

### 5.3. H7N9 (Avian Influenza with Limited Spillover)

Background: H7N9 influenza virus emerged in China in 2013 as a low-pathogenic virus in poultry that caused severe illness in humans. Over the next few years, it caused annual winter epidemics linked to live bird markets. By 2017, H7N9 had evolved into a highly pathogenic form in birds, leading to even larger outbreaks in poultry and more human cases in the fifth wave (2016–2017). Cumulatively, as noted, H7N9 infected over 1500 people and killed about 600 [56,57]. Its case fatality (~39%) in hospitalized patients was extremely high, though there may have been undetected mild cases. Importantly, H7N9 did not transmit easily between humans; most cases were isolated or in small family clusters at most. The Chinese government’s introduction of a poultry vaccination program in late 2017 dramatically reduced H7N9 incidence [80]. Since 2019, no new human cases have been reported, and H7N9 is presumed controlled at least for now.

H7N9 remains notable for pandemic planning because it showed that an avian virus could adapt well enough to infect humans in large numbers while still not achieving full human transmissibility. It is a reminder that the barrier between efficient zoonotic infection and sustained human spread is a critical tipping point.

Nomenclature Issues: Throughout the event, H7N9 was known simply as H7N9 or avian influenza H7N9. The names of concern were often phrased as “novel avian influenza A(H7N9) virus”. This highlighted it was from birds and novel, but after a few years, it was not novel anymore. All human cases were clearly linked to poultry exposure, so one might argue the name was fine. However, as years passed, people might have lost track that this was still a bird virus and not a human virus. It also co-existed with H5N1 in some regions, leading to media confusion at times (two different “bird flu” threats). If H7N9 had ever started spreading in people, we again would have had to adjust how we talk about it.

#### How a Transmission-Based Name Helps

Initially, and indeed, throughout its known course, H7N9-Avian would be the designation. This asserts that birds (specifically, chickens in live markets) were the reservoir sustaining it. Public health messaging can then emphasize controlling the virus in poultry to stop human cases. If evidence had emerged of sustained human transmission (e.g., a sizable cluster not linked to a market, or community spread), an update to H7N9-Human would signal a dire change. That never happened, so a change in nomenclature was unnecessary. In a sense, the name staying as H7N9-Avian communicates that despite numerous cases, the virus never established a foothold in humans. It always required the avian source, remembering the risk.

One could consider whether H7N9 needed any temporal or geographic marker in its name. Since it was essentially one protracted outbreak (all in China, 2013–2018 [81]) by a particular lineage, an extra label was not really necessary. Had it spread to other countries significantly, perhaps something like H7N9-Avian-China could be used, but we prefer to avoid geographic terms. The year of emergence (2013) could be included as H7N9-Avian-2013 to distinguish it from any other H7N9 that might exist elsewhere (there are other avian H7N9 strains in birds globally, but they have not infected humans). However, to keep names short, it might be fine to omit the year unless needed for clarity.

In summary, H7N9’s case reinforces that the absence of a host update in naming (remaining “-Avian”) is in itself informative under our system. It means public health interventions should focus on the animal source, and that human-to-human spread has not become a feature of that virus. This stable naming until a real change occurs avoids false alarms. It also suggests a broader point: not every zoonotic virus will get a new label unless it truly shifts its transmission mode. This conservatism ensures that the nomenclature, while dynamic, does not fluctuate unnecessarily or cause confusion with too-frequent changes.

These case studies collectively show that a transmission-based nomenclature is feasible and is advantageous. In each instance, it either clarifies the situation retrospectively or would have improved communication during the event. H1N1pdm09 shows how quickly a name might need to change (within weeks) when a species jump happens. H5N1 and H7N9 show slower-developing situations, where a name change might never occur for H7N9, or only now be on the horizon after decades for H5N1, but having criteria and a plan for if/when it does would greatly aid preparedness. In the next section, we discuss the broader implications of adopting transmission-based nomenclature beyond these individual cases.

## 6. Implications of Transmission-Based Nomenclature

Adopting a transmission dynamics-based naming system for influenza viruses has implications for science, public health, and policy. It could influence how we prioritize research, how vaccines are developed, and how the public perceives different influenza threats. In this section, we explore key areas where a transmission-based nomenclature could add value or change current practices, specifically in enhancing vaccine development and improving public health communication. Later, in Section 7, we present the implications for global surveillance and multidisciplinary implementation strategies.

### 6.1. Enhancing Vaccine Development

Influenza vaccine strain selection involves anticipating strains that will be predominant in the upcoming season or could cause a pandemic [82,83,84]. A naming system that highlights which host a virus is adapted to can provide guidance for these decisions. For example, if a virus is labeled “-Human”, that flags it as a current threat to humans and a candidate for inclusion in vaccines or for pandemic stockpile efforts. Conversely, a virus labeled “-Avian” or “-Swine” might not warrant a human vaccine unless it shows signs of moving toward humans.

Take H5N1: If, one day, we have an H5N1-Human strain, it would be obvious that developing or updating an H5 vaccine for humans becomes urgent. In the current system, H5N1 is already recognized as a potential pandemic strain, but the nomenclature does not differentiate between the H5N1 causing poultry outbreaks and a hypothetical H5N1 that has adapted to people. Transmission-based naming would make that distinction explicit and immediate.

Another benefit is streamlining communication in the vaccine community. Researchers working on “H7N9-Avian” know they are dealing with a zoonotic strain that might require a different approach (e.g., stockpiling vaccines but not deploying them widely unless it shifts). If it were to become “H7N9-Human”, it might be incorporated into seasonal flu vaccines in affected regions or trigger mass immunization campaigns. The name itself encapsulates the risk level. This could also help with regulatory and funding decisions—for instance, releasing funds for vaccine development might be easier to justify when the virus carries a “-Human” designation (indicating a clear and present human risk).

During the 2009 H1N1 pandemic, one challenge was that the virus emerged and spread so rapidly that by the time a vaccine was produced, the pandemic had peaked in many places [85,86]. Part of the delay in recognizing the scope was initial confusion; if an official label “H1N1-Human-2009” had been applied as soon as sustained transmission was verified, it might have more quickly signaled vaccine manufacturers to switch gears from seasonal to pandemic production. In practice, they did so by June 2009 [87], but clear nomenclature can only help in removing ambiguity.

Moreover, for seasonal vaccine updates, consider a scenario with multiple co-circulating strains of the same subtype. Suppose there are two H3N2 influenza strains, an H3N2-Human (current strain) and an H3N2-Swine that occasionally infects humans (variant). If the swine-derived one starts increasing in prevalence in humans (but still not sustained), our naming might temporarily label isolates from humans as H3N2-Variant or similar. If it crosses the R_0_ > 1 threshold to human adaptation, it becomes H3N2-Human-20XX. Such clarity informs vaccine strain selection committees that a new human-adapted lineage has joined the mix and might need representation in the vaccine if it competes with the older lineage.

In summary, transmission-based nomenclature is heuristic for vaccine targeting. It identifies which virus lineages are in which hosts, thereby guiding the strategic focus of vaccine research and stockpiling. It complements genetic characterization with an extra layer of phenotypic relevance, essentially differentiating “is this virus currently a human problem or an animal problem?”—a key question for vaccine strategy.

### 6.2. Improving Public Health Communication

Perhaps one of the strongest arguments for transmission-based naming is the benefit to communication and education. When an outbreak occurs, public ignorance or understanding of what is happening can influence everything from personal protective behaviors to acceptance of control measures. A naming system that inherently reduces confusion and conveys risk can improve the effectiveness of communication.

1. Reducing Stigma and Panic: By avoiding geographic names, the proposed nomenclature prevents the unfair blaming of regions or ethnic groups for the emergence of a virus. It focuses attention on the true source of risk, which is the infected host species and the virus’s mode of spread. This can make public messaging more factual. For instance, telling people, “H5N1-Avian is spreading in our poultry—avoid contact with sick birds” is more precise than “bird flu is here”, which might cause people to fear wild birds indiscriminately or, conversely, not realize their backyard chickens are a risk. Similarly, when COVID-19 variants were named with Greek letters instead of countries [88], it helped shift discourse away from finger-pointing and toward the characteristics of the variants. We could see a parallel in flu: no more “Spanish flu” or “Asian flu” terminology revival, but instead, names like “H1N1-Human-1918” in historical discussion, etc., which are neutral.

2. Clarifying Transmission Risk: A dynamic name would convey when a virus has become more dangerous to humans. If the public hears that an “avian influenza” has changed to a “human influenza” in naming, that inherently warns them that person-to-person spread is now happening, and thus they might need to take precautions (like mask use, avoiding crowds, etc., as advised by health authorities). Conversely, as long as it is “avian influenza H7N9”, an informed public understands that preventing infection involves avoiding poultry exposure, and not catching it from another human not exposed to live poultry, thus preventing overreaction or lack of reaction. During the H7N9 outbreaks, there was fear worldwide that it could spark a pandemic, but there was also knowledge that it was not spreading in communities [89]. Clear naming could reinforce that: e.g., health agencies could repeatedly refer to it as “H7N9 avian virus” to remind people of the limited transmission. If, in a hypothetical scenario, it became “H7N9 human virus”, that shift in terminology would be a clear alert.

3. Media and Public Discourse: The media often struggles with scientific nuance. They might report, “scientists fear bird flu may start a pandemic”. If, instead, the official language classifies viruses as not (yet) pandemic-capable vs. capable, then reporters have a straightforward term to use. They could say, “Currently H5N1 is labeled an avian virus, meaning it does not spread among people. Officials are watching closely in case it shifts to a human-adapted virus”. This is easier to follow. It also might reduce sensationalism, because the terms are somewhat technical but still understandable. The difference between a zoonotic threat and a human epidemic would be baked into the name, and thus, into reporting. During crises, consistent terminology is crucial—e.g., the shift to calling 2009 H1N1 “pandemic (H1N1) 2009” created consistency in documents. A transmission-based scheme would similarly provide a consistent vocabulary, with terms like “Human-adapted flu strain” directly in the name.

4. Education and Awareness: Over time, if this nomenclature is adopted, the public may become more literate in the concepts of zoonosis and host jumps [90,91]. People would become used to hearing things like “mammalian-adapted flu” or “avian-origin flu” and understand the difference. This is analogous to how terms like “variant” or “strain” became commonly understood during COVID-19. As a result, it might be easier to explain why certain measures (culling chickens, vaccinating pigs, closing markets) are needed for an “avian” virus versus why travel restrictions or social distancing might be needed if it becomes a “human” virus. The name itself sets the context.

Overall, tying nomenclature to transmission dynamics yields dynamic communication: it changes when the situation changes, ensuring that the name of the virus always aligns with the message health authorities need to convey. This stands in contrast to static names that require footnotes and explanations as the situation evolves (e.g., “Yes, it is called swine flu, but you can get it from people directly now, not pigs, …”). By reducing those explanatory gaps, we can deliver information more efficiently and improve public compliance with recommendations. As noted by one commentary, clear terminology is a critical part of public health transparency and trust.

Having considered the advantages in specific domains like vaccines and communication, we next turn to the practical aspects of making this nomenclature system a reality. In the final sections, we discuss what changes in surveillance, policy, and collaboration would be required to implement transmission-based naming on a global scale, and provide recommendations for next steps.

## 7. Integration into Global Public Health and Surveillance Systems

Implementing a transmission-based influenza nomenclature in practice will require coordinated efforts across the scientific, public health, and policy domains. Here, we outline strategies and recommendations for integrating this naming framework into global surveillance systems and pandemic preparedness activities. Key considerations include establishing clear criteria for renaming, ensuring transparency of data, achieving international consensus, and fostering cross-disciplinary collaboration (the One Health approach) to inform naming decisions. By proactively developing the infrastructure and agreements needed, the global community can be better prepared to adopt transmission-based names that improve pandemic response.

### 7.1. Strengthening Surveillance and Research to Inform Nomenclature

A transmission-focused naming system is only as good as the data that drive it. To determine when a virus has achieved sustained transmission in a new host (and thus merits a naming update), we need robust surveillance and research efforts.

#### 7.1.1. Enhanced One Health Surveillance

Influenza surveillance should explicitly integrate human, animal, and environmental health monitoring. Programs like the WHO’s Global Influenza Surveillance and Response System (GISRS) [92] and veterinary networks under FAO/OIE (now WOAH) [93] must share data in real time. Surveillance in poultry, swine, wild birds, and even farmed wildlife or companion animals can provide early warning of viruses adapting to new hosts. For example, upon detection of a jump from birds to mammals (as with H5N1 in mink or seals), veterinary authorities should alert public health officials immediately. Joint investigations can then assess if mammal-to-mammal transmission occurred. Surveillance protocols should define what evidence (e.g., multiple detections in the same species, genomic changes associated with adaptation, etc.) would trigger considering a host category change for the virus name. Global plans encompassing strengthening lab capacities to sequence and characterize viruses from different hosts are crucial as a coordinated One Health surveillance strategy.

#### 7.1.2. Transmission Studies and Thresholds

Research identifying the markers of sustained transmission is crucial. This includes field epidemiology (monitoring clusters and calculating R_0_ when possible) and laboratory studies (ferret transmission experiments, for example, to see if a virus can spread via respiratory droplets in a mammalian model). By correlating genetic changes with transmissibility data, scientists might predict when a virus is nearing a phenotype shift. These findings should feed into risk assessments. A multidisciplinary expert group (virologists, epidemiologists, veterinarians) must regularly review emerging influenza strains and advise whether to trigger a nomenclature update. For instance, if unusual clusters of H7N9 without bird contact had appeared, this group would evaluate if criteria for inserting the “Human” designation to the viral name had been met. The establishment of formal criteria or a decision algorithm is necessary—for example: “If R_0_ is estimated > 1 in new host or there is confirmed sustained transmission beyond 2 generations in non-reservoir host, then update host label”. These criteria, if internationally recognized, naming decisions are then transparent and backed by scientific evidence.

#### 7.1.3. Data Sharing and Transparency

Open data sharing is fundamental. Countries should promptly share virus sequence data and epidemiological findings on platforms like GISAID and report any unusual transmission events through WHO and OIE channels. The naming framework itself could incentivize transparency since it avoids country names in the nomenclature and thus encourages countries to report. An emphasis on transparency is paramount for international cooperation. When surveillance data are openly available, the global scientific community can help detect patterns that indicate a host adaptation. For example, multiple labs analyzing H5N1 sequences from disparate mammal cases might independently notice the same adaptation markers, collectively building the case for a naming update.

#### 7.1.4. Historical Analysis and Baseline

Researchers should retrospectively apply the framework to past cases (as partly done in this paper) to develop a baseline understanding. By simulating how we would have renamed past viruses, we can test the logic and fine-tune thresholds. This also would help train current epidemiologists in the new thinking. For instance, analyze the 1918, 1957, 1968, 1977, and 2009 events: at what point in those years would surveillance data have justified switching to the “Human” label? Doing so might reveal that in 1918 it would have been perhaps too late (since surveillance was poor), whereas in 2009 it could have been flagged in early May 2009 from outbreak investigations. These lessons can calibrate our present-day criteria.

In summary, to implement naming changes swiftly and appropriately, we need a strong evidence base. Investments in surveillance and research—especially in detecting cross-species transmission events—are investments in better naming and thus better warning systems [94,95]. As one FAO strategy document notes, a One Health approach with integrated data is key to contextualizing and responding to influenza threats in a globalized world. With such systems in place, the decision to label a virus “-Human” or “-Mammal” would come as a natural conclusion of the data, not a subjective or controversial call.

### 7.2. Fostering Transparency, Coordination, and Collaboration

Implementing transmission-based nomenclature globally will require consensus and cooperation among international bodies, national governments, and scientific experts. Several steps can facilitate this.

#### 7.2.1. International Endorsement and Guidelines

The World Health Organization (WHO), in collaboration with the World Organization for Animal Health (WOAH, formerly OIE) and the Food and Agriculture Organization (FAO), should convene an expert working group to formalize the nomenclature scheme. These organizations have the legitimacy to set naming standards, much as WHO has done for COVID-19 variants and for past influenza nomenclature decisions. The working group can develop a guidance document that defines the naming format, criteria for updates, and procedures for announcing a name change. Having WHO/FAO/WOAH jointly issue this guidance will embed One Health principles and ensure both human and animal health sectors are integrated. The document should also encourage countries to adopt the terminology in their reporting and publications. This top-down endorsement is crucial to achieving uniform usage; otherwise, disparate names could proliferate in different regions.

#### 7.2.2. Coordination Mechanism

We propose establishing a Nomenclature Coordination Committee possibly under the existing WHO framework (e.g., as part of the GISRS or as an adjunct to the WHO flu strain selection committee). This committee would review data and decide on nomenclature changes in real time. It would include members from WHO, WOAH, FAO, leading researchers, and representatives from heavily affected countries. If surveillance data suggest a virus has met the criteria for a host-label change, the committee can rapidly confer (virtually if needed) and make a determination. Once a decision is made, WHO and WOAH can simultaneously issue alerts to inform all member states that, for example, “Influenza A H5N1 clade X is now being designated as H5N1-Mammal to reflect documented mammalian transmission”, providing a summary of evidence [96]. This ensures that communication of change occurs through official channels, reaching national influenza centers, veterinary authorities, and ministries of health together. Such coordination was exemplified in 2011 when WHO consulted with partners to standardize the term A(H1N1)pdm09; a similar collaborative approach can be used for dynamic naming.

#### 7.2.3. National Adoption and Public Health Integration

Countries should integrate the new nomenclature into their pandemic plans and communication strategies. National influenza centers and outbreak response teams could include in their standard operating procedures an item to check WHO updates on nomenclature. If a virus of concern is reclassified, national authorities should update public messaging, guidelines, and possibly policy measures accordingly. For example, an animal outbreak that leads to a “-Mammal” classification might trigger a country to activate certain preparedness steps (like stepping up PPE for animal outbreak responders, reviewing vaccine seed strains, etc., analogous to moving to a higher phase in WHO’s pandemic alert system). The naming system, in effect, can complement or enhance existing pandemic phase alerts by providing a more detailed, virus-specific alert. National adoption also means training health communicators to use the terms correctly, so that press releases and risk communications align with the nomenclature.

#### 7.2.4. Cross-Disciplinary Collaboration

The very nature of transmission-based naming encourages collaboration between virologists, epidemiologists, veterinarians, and ecologists. To decide what a virus is “demonstrating” in terms of transmission requires cross-disciplinary inputs from all expertise. We recommend more joint analyses and publications across sectors. For instance, after a major spillover event, a One Health investigation team can publish a report that not only describes the outbreak but also explicitly assesses whether criteria for sustained transmission were met [97]. This team approach is obvious in some instances (e.g., joint human–animal investigations of H7N9 markers show that it is feasible). Institutionalizing such collaborations will yield data with more coherence for naming decisions. The One Health High-Level Expert Panel (OHHLEP) stresses the importance of communication, coordination, collaboration, and capacity (the “4Cs”) in managing zoonotic threats. Applying those same 4Cs here would ensure constant communication between human and animal health sectors about viral spread; coordinate on what the data mean; collaborate on the response, including what to call the virus; and build capacity to do all this in every country.

#### 7.2.5. Transparency in Decision Making

Full transparency backing scientific rationale (data on cases, R_0_ estimates, sequences) is obligatory alongside the announcement. This allows the broader scientific community to understand and, if needed, debate or validate the decision. Transparent criteria and evidence help maintain trust. Just as WHO publishes vaccine strain selection reports, naming decisions are available in outlets like the Weekly Epidemiological Record or disease outbreak news. By being open, we also invite independent researchers to potentially spot things the official channels might miss (extra sets of eyes on the data).

Implementing these coordination and transparency measures will ensure that transmission-based nomenclature is not just an academic idea, but a practical tool ingrained in the global influenza governance structure. The ultimate aim is that when the next potential pandemic virus emerges, the world will have a naming system that immediately reflects its status and risk, enabling faster consensus on actions. As one *Lancet* commentary noted, building global preparedness for avian influenza requires strong international coordination and communication [96]—naming conventions should be part of that preparedness.

Finally, while focusing on influenza, it is worth noting that the principles here could extend to other emerging viruses. The COVID-19 experience taught us the value of clear variant naming. In the future, perhaps even novel pathogens might have naming schemes that incorporate transmission mode or reservoir (imagine if “2019-nCoV” had been called “Airborne-CoV-2019” once it was known to be primarily airborne human-to-human—it might have signaled the response needed). Influenza can lead the way in this innovation, given its long history of problematic naming and the rich surveillance data we already have.

## 8. Conclusions

Influenza remains an enduring global health threat, and with each emergence of a novel strain, our capacity to respond swiftly and effectively is a constant challenge. Among the many tools available to enhance preparedness and risk communication, nomenclature is often underestimated. Yet, the names we assign to viruses are not neutral—they shape public perception, guide political and economic decision-making, and influence the speed and coherence of international response.

This article proposes a transmission dynamics-based nomenclature framework that addresses the critical limitations in current naming practices, which often rely on static attributes such as hemagglutinin/neuraminidase subtype, host of first isolation, or geographic labels. While these classifications are valuable for virologists, they frequently fail to convey the most urgent epidemiological information, i.e., the current dominant host and whether the virus is demonstrating sustained transmission (e.g., R_0_ > 1) in that host.

By analyzing both historical and contemporary cases—including 1918 H1N1, 1957 H2N2, 1968 H3N2, 2009 H1N1pdm09, H5N1, and H7N9—we demonstrate that adopting names that evolve with host shifts (e.g., H5N1-Avian → H5N1-Mammal) could improve situational awareness. This dynamic, host-tagged nomenclature would complement the existing genomic identifiers by integrating real-time epidemiological risk directly into the naming structure.

The proposed framework is conservative and evidence-based, triggering name changes only under specific conditions:Documented sustained transmission (R_0_ > 1) in a new host species;Recurrent outbreaks in that host across geographically distinct populations;Serological or genomic evidence of population-level adaptation or spread.

Our proposal maintains compatibility with scientific nomenclature but enhances its communicative power by embedding host ecology and transmission dynamics into the name itself. This enables clearer, timelier messaging during outbreaks, reduces the use of stigmatizing geographic labels, and signals to vaccine developers and policymakers when a virus is entering a new phase of risk.

Of course, challenges remain:Determining the exact point at which a virus has achieved “sustained transmission” in a new host can be difficult, especially with limited or lagging data.Frequent or premature name changes could generate confusion if not accompanied by coordinated guidance.Public understanding of such naming conventions must be empirically studied to ensure that the benefits of clarity outweigh any unintended misperceptions.

To mitigate these risks, we recommend the following:Establishing predefined, internationally agreed thresholds for renaming events;Piloting the system during future zoonotic spillovers of interest (e.g., H5N1 mammal outbreaks);Integrating the framework into One Health-oriented pandemic preparedness simulations.

In conclusion, a dynamic, transmission-informed nomenclature system is a logical and necessary evolution reflecting the increasing importance of real-time data and host ecology in managing influenza threats. The virus’s name should evolve as its behavior does.

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
