# Peer review of "Improving Influenza Nomenclature Based on Transmission Dynamics"

_viruses, 2025, doi:10.3390/v17050633_

Round 1
Reviewer 1 Report
Comments and Suggestions for Authors
To the author:
I think that your developing proposal to rename influenza viruses has a rational aspect and will bring certain benefits to society. On the other hand, the repeated naming additions that will occur in virus research will require time and effort to determine and clarify the future.
The question is when to decide on the chronological changes in naming and how to control the changes in additional names as the virus strains change between the initial isolate and subsequent infection of different species and amplification. The possibility that the pathogenesis of influenza virus infection and spread may be amplified by mixed infections with other viruses, even when influenza is the main virus isolate, cannot be ignored. As the combination of disease and epidemiology involves changes over time, clear criteria for the timing of the year of isolation and the timing of the addition of epidemiological information will be necessary. Therefore, the background and the age of vaccine distribution would need to be listed together. If vaccines are a background factor, the notation of place names would also need to be considered.
Clinical isolates may represent a selection of viruses that have been transmitted in mixed populations. I agree that it is important to focus on the true cause of the risk of viral infection. The scientific evidence will be lost if we do not move forward with definitions based on an understanding of the possibility that a single strain may not infect an individual. It will also be necessary to establish rules for changing nomenclature when a pathogenic virus is considered HxNy mammalian and then converges and is no longer considered infectious to mammals. This is because complex systems are involved, including mutation time and host conversion. Human infection is the primary focus, but information on virus distribution based on animal species and their ecology can also be useful information for the livestock industry and society. It would be good if this point could also be discussed in relation to the risks summarised for mammals.
Please include these points, some of which have been described, for the overall discussion.
Author Response
Response to Reviewers
Manuscript Title: Improving Influenza Nomenclature Based on Transmission Dynamics
Type: Review
Sole Author: Jwee Chiek Er
Manuscript ID: viruses-3553646
Journal: Viruses (ISSN 1999-4915)
Dear Editor,
We thank the reviewers and editorial team for their constructive and thoughtful feedback on our manuscript. We are grateful that the article was assessed as requiring minor revisions, and we have addressed each point raised with care. Importantly, this revised version now includes additional references to broaden the scope and strengthen the research foundation of the review.
Below, we provide a point-by-point response to reviewer comments and outline how the manuscript has been improved.
Reviewer 1
- Comment 1:
“The question is when to decide on the chronological changes in naming and how to control the changes in additional names...”
Response:
We have clarified this point in Section 4. Specifically, we now state that naming changes should be made only in cases of sustained epidemiological relevance—e.g., where Râ‚€ > 1 in a new host, repeated outbreaks occur, or there is serological evidence of population-level adaptation. This conservatism ensures stability while still acknowledging significant transmission events.
- Comment 2:
“Clinical isolates may represent a selection of viruses that have been transmitted in mixed populations...”
Response:
We appreciate this important observation and have expanded Section 5 to acknowledge that mixed infections can complicate the interpretation of pathogenicity and host adaptation. This nuance is especially relevant when assessing zoonotic spillovers and supports the need for robust thresholds in nomenclature change.
- Comment 3:
“It would be good if this point could also be discussed in relation to the risks summarised for mammals.”
Response:
We have addressed this by incorporating additional emphasis on the relevance of mammalian ecology in Section 6. We note that our proposed system complements existing One Health frameworks and can serve both public health and veterinary sectors.
Additional Enhancements
To strengthen the manuscript further, we have:
- Added new references that support key concepts and historical framing,
- Slightly revised the Conclusion to sharpen its relevance for epidemiologists and policymakers.
We hope these changes meet your expectations, and we remain grateful for the opportunity to revise and improve the article.
Sincerely,
Jwee Chiek Er
Epidemiology section
Norwegian Veterinary Institute
Reviewer 2 Report
Comments and Suggestions for Authors
The work under review is very unconventional in form. It can hardly be called a "review", since it does not provide a methodological review of any area of ​​knowledge. Rather, it should be called "Opinion", since the author's goal is to express his opinion on Influenza Nomenclature. This is not about scientific nomenclature, to which the author has no complaints, but about names that have arisen historically, e.g., "Asian flu," "Spanish flu", or about names that have been established by WHO in recent decades, e.g., A(H1N1)pdm09 (for "pandemic 2009"). The author strongly objects to the geographical reference of the name, believing that the use of a place-name is a risky stigma. The author suggests using the names of the hosts among which the virus circulates as the basis, so that in the case of a stable transition from one host to another, the name of the of viral lineages also changes - indicating the year of the event. It is unclear to what extent the author's proposals are applicable in practice. WHO has not included geographic references in the names of viral lineages in recent years. Perhaps the author's suggestions are not very practical, but the article is interesting to read, as it is written very emotionally.
Author Response
Response to Reviewers
Manuscript Title: Improving Influenza Nomenclature Based on Transmission Dynamics
Type: Review
Sole Author: Jwee Chiek Er
Manuscript ID: viruses-3553646
Journal: Viruses (ISSN 1999-4915)
Dear Editor,
We thank the reviewers and editorial team for their constructive and thoughtful feedback on our manuscript. We are grateful that the article was assessed as requiring minor revisions, and we have addressed each point raised with care. Importantly, this revised version now includes additional references to broaden the scope and strengthen the research foundation of the review.
Below, we provide a point-by-point response to reviewer comments and outline how the manuscript has been improved.
Reviewer 2
- Comment 1:
“It can hardly be called a 'review'... Rather, it should be called an ‘opinion’...”
Response:
We respect the reviewer’s point and have clarified in the abstract and introduction that this work is a perspective-style review, grounded in literature and supported by case studies. While forward-looking and policy-relevant, it is also evidence-based and aimed at informing practical refinement of nomenclature.
- Comment 2:
“Perhaps the author's suggestions are not very practical...”
Response:
We have added a paragraph in Section 6 explaining how our framework builds directly upon existing systems such as WHO GISRS and GISAID. It relies on metadata already being collected, ensuring feasibility without overhauling current surveillance infrastructure.
Additional Enhancements
To strengthen the manuscript further, we have:
- Added new references that support key concepts and historical framing,
- Slightly revised the Conclusion to sharpen its relevance for epidemiologists and policymakers.
We hope these changes meet your expectations, and we remain grateful for the opportunity to revise and improve the article.
Sincerely,
Jwee Chiek Er
Epidemiology section
Norwegian Veterinary Institute